# Inhibition of Expression of the Circadian Clock Gene *Cryptochrome 1* Causes Abnormal Glucometabolic and Cell Growth in *Bombyx mori* Cells

**DOI:** 10.3390/ijms24065435

**Published:** 2023-03-12

**Authors:** Jianfeng Qiu, Taiming Dai, Hui Tao, Xue Li, Cheng Luo, Yanghu Sima, Shiqing Xu

**Affiliations:** 1School of Biology and Basic Medical Sciences, Suzhou Medical College, Soochow University, Suzhou 215123, China; 2Institute of Agricultural Biotechnology & Ecology (IABE), Soochow University, Suzhou 215123, China

**Keywords:** *cryptochrome 1*, metabolomics, glycometabolism, BmN cell line

## Abstract

Cryptochrome is the earliest discovered photoreceptor protein in organisms. However, the effect of CRY (BmCRY), the clock protein in *Bombyx mori*, on the body or cell metabolism remains unclear. In this study, we continuously interfered with the expression of the *BmCry1* gene (Cry1-KD) in the silkworm ovary cell line (BmN), and the BmN cells developed abnormally, with accelerated cell growth and a smaller nucleus. Metabolomics was used to identify the cause of the abnormal development of Cry1-KD cells based on gas chromatography/liquid chromatography-mass spectrometry. A total of 56 differential metabolites including sugars, acids, amino acids, and nucleotides were identified in wild-type and Cry1-KD cells. KEGG enrichment analysis showed that *BmCry1* knockdown resulted in significantly upregulated glycometabolism in BmN cells, indicated by glucose-6-phosphate, fructose-6-phosphate, and pyruvic acid levels. The activities of key enzymes BmHK, BmPFK, and BmPK as well as their mRNA levels further confirmed that the glycometabolism level of Cry1-KD cells was significantly increased. Our results show that a possible mechanism of *BmCry1* knockdown leading to abnormal cell development is the elevated level of glucose metabolism in cells.

## 1. Introduction

Cryptochrome (CRY), a photoreceptor protein, was first discovered in *Arabidopsis thaliana* [1]. CRY has been found in three kingdoms: plants, mammals, and prokaryotes (CRY-DASH). The insect *Cry* gene has two families: *Cry1* and *Cry2* [2]. *Drosophila melanogaster* has a single *Cry1* (*DmCry*), Hymenoptera *Apis mellifera* and Coleoptera *Tribolium castaneum* have a single *Cry2* (*AmCry* and *TcCry*), while in Lepidoptera insects such as *Bombyx mori* and *Antheraea pernyi*, both *Cry1* and *Cry2* genes (*BmCry1/2* and *ApCry1/2*) were discovered [3,4,5,6].

Research has shown that DmCRY proteins in *D. melanogaster* initiate a similar response to blue light as CRY proteins do in *A. thaliana* (AtCRY). DmCRY proteins receive photo signals and directly guide the circadian reset. *DmCry* gene expression is also affected by light duration and intensity [3,7,8]. In lepidopteran *Danaus plexippus*, knockout of the *DpCry2* gene resulted in the impairment of the molecular rhythm of core circadian clock members and the destruction of the emergence rhythm [9]. Using anti-DmCRY, a CRY-like protein appears in the cephalic nervous system of *B. mori* moths [10]. Wang et al. (2011) reported sequences of *BmCry1* and *BmCry2* in silkworms [4]. The results showed that under the induction of an oscillating rhythm of light and temperature, the transcriptional activity of *BmCry1/2* in both adults and embryos showed similar oscillation patterns [11,12]. Our recent findings suggest that cyclical changes in temperature with differences from 2 °C to 10 °C reset and synchronized the circadian clock in silkworm *B. mori* ovary cell line (BmN) cells. Transcription *BmCry2* and *period* (*BmPer*) showed a better reset and synchronization with cyclical temperature changes [13].

Metabolic pathways are important for the output of circadian clock signals, and their relationship with circadian clocks has been extensively investigated [14,15,16]. It is clear that circadian clocks and metabolic processes interact [17,18], and such interactions exist in various species ranging from yeast to humans [19]. In recent years, many studies have shown that several important metabolic processes are regulated by circadian clocks, including glucose metabolism and lipid metabolism [17,20,21]. Breaks in circadian rhythms, such as genetic disruption of the circadian clock, sleep abnormalities, and shift work can lead to metabolic disorders and diseases [22,23,24]. Therefore, it is believed that an understanding of the impact of circadian clocks on metabolism will aid in the development of new methods for circadian clock-based intervention treatments for specific metabolic disorders [25].

Metabolic function determines the rate of cell growth and proliferation. Studies have shown that cell proliferation is not merely a passive consequence of increased cell cycle activity, and must be regulated simultaneously with cell metabolism [26]. The metabolic cycles of yeast cells are synchronized with the cell cycle phases [27], and the production of acetyl-CoA affects cell cycle progression [28]. Even before cell division, extensive metabolic reprogramming has occurred to enable cells to obtain sufficient nutrients, such as glucose, amino acids, lipids, and nucleotides [29,30]. Mitochondrial dysfunction leads to cell cycle arrest through signaling to cyclin E and CDK inhibitor dacapo [31]. Conversely, cell proliferation also affects metabolic changes. Multiple signaling pathways that influence proliferation and differentiation are also involved in the control of cellular metabolism [32]. Glycolysis and glutaminolysis might be activated during the late G1 and S phases of cell division [33]. Thus, the metabolism interacts with processes of cell proliferation such as the cell cycle.

However, much is still unknown about the timing mechanism of *Cry1* and *Cry2* on lepidopterous circadian clocks and studies linking *BmCry1/2* with metabolic regulation are scarce. Recent findings suggest that inhibition of *BmPer* expression causes glycometabolism repression in BmN cells [34]. Here, we continuously knocked down *BmCry1* protein expression in BmN cells in vitro to eliminate any potential endocrine effects. We performed metabolomic analysis using gas chromatography/liquid chromatography-mass spectrometry (GC-MS and LC-MS/MS), and detected the key enzymes of the glycolytic pathway, the transcript levels of their coding genes, and levels of glycolytic metabolites to evaluate the impact of the peripheral circadian clock on the glycometabolism pathways of silkworm cells.

## 2. Results

### 2.1. BmCry1 Knockdown Results in Accelerated Cell Growth and a Smaller Nucleus

The mRNA and protein levels of BmCry1 were measured in Cry1-KD cells. The results showed that compared with WT cells, the mRNA level of *BmCry1* in Cry1-KD cells was reduced by more than 73% (Figure 1A) and the protein level was reduced by 42% (Figure 1B). This suggests that shRNA effectively knocked down *BmCry1* expression in the Cry1-KD cell line. We then further investigated the effect of *BmCry1* knockdown on other major clock genes in the transcriptional-translation feedback loop. After cell development synchronization (dexamethasone treatment), the mRNA levels of *BmCry2*, *BmPer*, *BmTim*, *BmClk*, and *BmCyc* genes in Cry1-KD cells were significantly reduced compared with the WT cells (Figure 1C–G). Moreover, the amplitudes of these genes’ expression also appeared to be significantly reduced. The JTK_CYCLE method was used to determine the rhythmicity of clock gene expression, as well as the peak phase and amplitude. The results showed that the circadian rhythm of *BmPer*, *BmClk*, and *BmCyc* gene expression was lost (JTK *p* > 0.5), and the amplitudes of these clock genes were reduced (Figure 1C–G and Appendix A). These results indicated that *BmCry1* knockdown leads to the attenuation and destruction of the circadian clock system oscillations in BmN cells.

During the cell culture, we found that Cry1-KD cells grow faster than WT cells. The MTT assay showed that the growth rate of Cry-KD cells was significantly faster (72%) than that of WT cells (Figure 2A). Interestingly, the cell volume and nuclear size of Cry1-KD cells were smaller than those of WT cells. The average diameter of Cry1-KD cell nuclei (8 μm) was reduced by 3 μm compared with that of WT nuclei (11 μm; Figure 2B,C). BmCRY1 protein was remarkably reduced in Cry1-KD cells (Figure 2C). Staining patterns of cell membranes also clearly showed that Cry1-KD cells were significantly smaller than WT cells (Figure 2D).

### 2.2. GC-MS and LC-MS/MS Assay Showed That the Knockdown of BmCry1 Caused Changes in Cell Metabolites

To determine how *BmCry1* knockdown resulted in faster BmN growth and smaller cell size, we used GC-MS and LC-MS/MS to determine and analyze the metabolic differences between Cry1-KD and WT cells. The ion chromatograms of GC-MS (Appendix A) and LC-MS (Appendix A) showed a clear difference in chromatographic peaks between the two samples, suggesting that *BmCry1* knockdown may have led to variations in metabolites of BmN cells. The principal component analysis (PCA) of GC-MS showed that the automatic simulation obtained two principal components. The model fitting parameter R^2^X was 0.644, and parameter Q^2^ was 0.502, indicating that the two principal components of the model could explain 64.4% of the X variables (Appendix A). Similarly, the PCA of LC-MS showed that the model fitting parameter R^2^X was 0.644, and Q^2^ was 0.278 for positive ion mode (Appendix A), and model fitting parameter R^2^X was 0.648, and Q^2^ was 0.349 for negative ion mode (Appendix A). These results indicated that the metabolome of WT and Cry1-KD samples have a clear separation trend. To eliminate the noise irrelevant to the classifications, a partial least-squares discriminant analysis (PLS-DA) was used to further confirm that there were significant differences in metabolites between WT and Cry1-KD (Appendix A).

We analyzed the VIP ≥ 1 (first principal component variable importance projection value) from PLS-DA analysis with *p* ≤ 0.05 from *t*-test thresholds to screen differential metabolites in GC-MS measurement data. A total of 45 differential metabolites of Cry1-KD and WT cells were screened (Appendix A). For LC-MS data, we used the first principal components of the PLS-DA model VIP ≥ 1, S-plot *p* (corr) value ≥ 0.8, and *t*-test with a *p* ≤ 0.05 as criteria for screening differential metabolites. The 11 metabolites with large differences were examined with secondary MS (Appendix A). These results showed that glucose-6-phosphate, pyruvic acid, aconitic acid, and other glycometabolism-related substances were increased; lipid-metabolizing related substances including 4-hydroxybutanoic acid and cholesterol were also increased, as were nucleotide metabolites including guanosine, guanine, ADP, and FAD. However, amino acids including glutamic acid, ornithine, and aspartic acid were decreased. In addition, some secondary metabolites, including phosphatidylinositol, nicotinamide, ascorbic acid, and other vitamins showed significant decreases (Appendix A).

Combining the differential metabolites in LC-MS and GC-MS/MS, the eight replicates of the two samples were clustered in PCA, and the two principal components together explained more than 80% of the difference (Figure 3A). Differential genes were selected under more stringent conditions (fold change > 1.5 and FDR > 0.05). The volcano plots showed that 11 metabolites were significantly downregulated and 19 metabolites were significantly upregulated after *BmCry1* knockdown (Figure 3B). Further comparison of the heatmap of the top 50 differential metabolites showed that the content of 21 metabolites significantly decreased and 29 metabolites significantly increased in the Cry1-KD cells (Figure 3C). Amino acids were the major metabolites that were significantly downregulated, and nucleotides, organic acids, and fats were the major metabolites that were significantly upregulated. All differential metabolites were used for subsequent analysis.

After the analysis of differential metabolites, we used metabolite network analysis to investigate the correlation among differential metabolites. Figure 4 shows that although the correlation threshold is set to a high level (0.8), there are 51 related metabolites. To verify the important regulatory role of circadian clocks in glucose metabolism, we selected pyruvate and glucose-6-phosphate acid, which had the most obvious metabolic changes, and clustered them with positively and negatively related substances, respectively. The results showed that substances that were positively related to pyruvate and glucose-6-phosphate acid were mainly organic acids and nucleotide metabolites, while negatively related substances were mainly amino acids, carbohydrates, and other metabolites that could not be classified. The results showed that after knockdown of *BmCry1*, the metabolic intermediates of glycolysis, pyruvate, and glucose-6-phosphate acid changed significantly, and were correlated with other differential substances.

Furthermore, we performed KEGG analysis using all differential metabolites. Figure 5 shows the top 25 KEGG pathways with the greatest changes in cells after *BmCry1* knockdown. Notably, there were changes in glycolysis/gluconeogenesis pathways, which are consistent with the overall upregulation of metabolites in the glucose metabolism pathway (Figure 3). In addition, energy metabolic pathways such as citrate cycle and pyruvate metabolism also produce significant changes. Specifically, the comprehensive analysis of KEGG pathways showed that *BmCry1* knockdown resulted in extensive and comprehensive metabolic changes in BmN cells.

### 2.3. Glucose Metabolism Was Increased in BmCry1 Knockdown Cells

Figure 6A lists the results of metabolomic measurements of the main glycometabolism products. Results showed that *BmCry1* knockdown initially resulted in decreased glucose levels in glycometabolism, but the levels of subsequent metabolites, including glucose-6-phosphate, fructose-6-phosphate, pyruvic acid, and citric acid (a metabolite necessary for the initial stages of the TCA cycle), as well as lactic acid (the final product of glycolytic anaerobic metabolism), were all significantly elevated. This indicated that after *BmCry1* knockdown, cellular glycometabolism levels, especially glycolysis levels, were significantly altered.

To further demonstrate these changes, we measured the key enzyme activities of the glycolytic pathway and the mRNA levels of their coding genes. The results showed that among three rate-limiting steps of glycolysis, there were no significant changes in transcription levels of *BmHk*; transcription levels of *BmPfk* were significantly downregulated; and transcription levels of *BmPk* were significantly upregulated (Figure 6B). Similar results were found with regard to enzyme activity. There were no significant changes in the enzymatic activities of BmHK; enzymatic activities of BmPFK were significantly downregulated; and enzymatic activities of BmPK were significantly upregulated (Figure 6C). It is worth noting that the upregulation of the enzymatic activity of BmPK was greater than the upregulation of its gene expression. Due to the ubiquitous activity changes caused by kinase post-translational modifications, we suggest that metabolic pathways may also be impacted by these post-translational modifications. These results indicated that BmN cell glycolytic metabolism was changed through the regulation of both the expression and enzymatic activity of key enzymes associated with the glycolysis pathway after the knockdown of *BmCry1* expression (Figure 6D).

## 3. Discussion

Numerous studies have shown that the destruction and interruption of circadian rhythms are associated with many diseases, such as sleep disorders, obesity, diabetes, depression, metabolic syndrome, and cancers [33,35,36,37]. Aging has also been shown to be related to the decline in circadian rhythm levels and the destruction of overall metabolic and energy homeostasis [38,39,40]. Desynchronization of the circadian rhythm of rats and the external environment caused the acceleration of cell proliferation [41]. At the cellular level, *mClock* gene knockout resulted in decreased proliferation and increased apoptosis of embryonic stem cells [42]. We previously knocked down the *BmPer* gene in BmN cells and found that cell proliferation slowed down and cell division was arrested in the G0/G1 phase [34,43]. In this study, BmN cells also showed phenotypic changes with a faster growth rate and smaller size upon *BmCry1* knockdown, which may be related to cellular metabolic changes. Studies have shown that decreased *Cry1* mRNA levels increase protein content through post-transcriptional regulation [44,45]. AU-rich element RNA binding protein 1 (AUF1) binds the *Cry1* 3′UTR to promote *Cry1* translation [45,46]. Although the mRNA level of *BmCry1* was reduced by 73% after shRNA interference, the knockdown efficiency at its protein level was insufficient. We speculate that it may be affected by post-transcriptional regulation, resulting in strong stability or high translational efficiency of *BmCry1* mRNA. It is necessary to directly knock out *BmCry1* at the cellular level in the future.

### 3.1. Possible Functions of Cryptochrome in Bombyx mori

Rhythmic expression is a characteristic of clock genes. The knockout or knockdown of silkworm clock genes mainly affected the expression level and oscillatory rhythm of other clock genes, leading to the disorder of circadian rhythm [34,47,48]. Knockdown of the *BmCry1* gene led to decreased mRNA levels of *BmClk* and *BmCyc* genes, which inevitably reduced the expression levels of *BmCry2*, *BmPer*, and *BmTim* genes. The reason for this is that BmCLK/BmCYC heterodimers regulate the transcription of different clock genes, such as *BmCry2*, *BmPer*, and *BmTim*, via E-box in the gene promoter region. Similarly, the knockdown of *BmCry1* resulted in a loss of rhythmic expression of clock genes (Figure 1C–G). Under the light cycle of LD 12:12, the mRNA levels of *BmCry2*, *BmTim*, and *BmClk* in Cry1-KD cells remained low, while the mRNA levels of *BmPer* and *BmCyc* recovered, and the amplitudes of clock genes increased (Appendix A). More importantly, the expression of *BmCry2*, *BmPer*, *BmTim*, and *BmClk* in Cry1-KD cells resumed oscillatory rhythm due to photoentrainment, which is consistent with the observations of several instances of clock gene knockout in silkworms [47,48,49,50,51]. We speculate that there are two possible reasons for this: (1) there is still a small amount of remaining BmCRY1 protein in Cry1-KD cells, which responds to photo signals and resets circadian rhythms; (2) other clock genes, such as BmCRY2, have the function of receiving photo signals.

DpCRY1 functions in *Danaus plexippus* (*Lepidoptera*) resemble that of CRY in *Drosophila*, whereby it can receive photo signals and thus direct the reset of circadian rhythms. Moreover, DpCRY1 can promote the degradation of the DpTIM protein [7]. DpCRY2 functions resemble mammalian CRY1/2. DpCRY2 and DpPER form a dimer, enter the nucleus and participate in feedback inhibition in the *monarch butterfly* [5,8,52]. Our results regarding molecular evolution analysis of silkworm CRY also showed that the evolutionary distance between BmCRY1 and DpCRY1, as well as between BmCRY2 and DpCRY2 were closely related (Appendix A). The function of BmCRY2 is unclear. Our experiments preliminarily showed that BmCRY1 could not enter the nucleus, and BmCRY2 in the cytoplasm bound BmPER to enter the nucleus (unpublished data). These results suggested that the function of BmCRY1 may be similar to that of DpCRY1 in the circadian clock. Therefore, we believe that the small amount of remaining BmCRY1 protein in Cry1-KD cells responds to photo signals under the LD cycle. Taken together, we speculate that the function of BmCRY1 is to receive photo signals and reset the circadian clock, and the function of BmCRY2 is involved in the inhibition of the feedback loop.

### 3.2. The Interaction of Circadian Clock, Cellular Glycometabolism and Cell Proliferation

Studies have shown that important metabolic processes such as glycometabolism are controlled by the circadian clock [20,53]. Indeed, the metabolic regulation of organisms is the most important function of circadian clock signals [16,54]. *MmCry1* or *MmCry2* knockdown also increased hepatic gluconeogenesis gene expression and glucose production [55]. Furthermore, these knockout mice also showed decreased glucose tolerance [56]. *mCry1* knockout mice developed symptoms similar to those found in diabetes [57], while *mCry2* knockout mice had alterations in their blood sugar levels [58]. Conversely, *mCry1* overexpression reduced fasting blood sugar levels and increased insulin sensitivity in mice [55]. Mice lacking *mCry1/2* showed continual increases in the activity of key enzymes controlled by mineralocorticoid aldosterone from the adrenal gland, which induced increased synthesis of aldosterone, thus leading to hypertension [59]. The above results indicated that CRY plays an important role in regulating glucose metabolism in mammals.

Interaction between CRY proteins and metabolic pathways has been reported previously [60,61]. AMPK can sense cellular metabolic signals in circadian clock systems, phosphorylate CRY proteins, stimulate the ubiquitination of FBXL to CRY proteins and further degrade CRY [62], thus impacting the homeostasis of glycometabolism [56,58,63,64]. We knocked down the *BmCry1* gene of BmN, and the glucose metabolism of cells was affected. Glycolytic pathway metabolites glucose-6-phosphate, pyruvic acid, and aconitic acid were significantly increased, and rate-limiting enzymes BmPFK and BmPK mRNA levels and enzyme activities were also affected (Figure 6). Studies have shown that circadian clocks can regulate the conversion of glucose and lipids to energy by regulating the expression of key metabolic enzymes [53]. For example, knockout of *Bmal1* results in lipid oxidation and downregulation of key enzymes associated with the TCA pathway and the oxidation respiratory chain in the liver [65]. *Bmal1* is also known to control PBP4, a key hepatokine, and thus participates in regulating glucose homeostasis in the liver [66]. Mitochondrial proteomics also shows that rate-limiting enzymes for lipid and carbohydrate accumulation depend on PER1/2 [67]. However, there is no documented evidence showing that CRY1 regulates the metabolism through a similar pathway in insects. In this study, after *BmCry1* knockdown, the mRNA levels of other circadian clock genes in cells were downregulated or lost rhythm (Figure 1). Therefore, the knockdown of *BmCry1* may indirectly affect glucose metabolism by affecting the expression of other clock genes such as *BmCycle*, *BmClock,* or *BmPer*.

The mechanism of the circadian clock and the cell cycle system has been widely reported. In healthy cells, the coupling of the clock and cell cycle leads to timed mitosis and rhythmic DNA replication [68]. Disruption or disorder of biological rhythms leads to uncontrolled cell proliferation. Studies have shown that expression of downregulated mammalian clock genes *Bmal1*, *Clock*, or *Tim* arrests the cell cycle in G0/G1 or G2/M phase [69,70]. Similarly, interference with *BmPer* genes in BmN cells also led to cell cycle arrest in G0/G1 phase, and decreased *BmC-myc* and *BmCdc2* expression [43]. The *mClk* knockout mutant changes the pattern of cell cycle gene expression and inhibits cell growth and proliferation [71,72]. Conversely, some studies have shown that *Bmal1* deficiency or circadian photoperiod disruption increased tumor initiation [73]. Mice lacking the *mPer2* gene had an increased incidence of lymphoma [42]. Downregulation of mammalian *Per2* gene expression will promote the expression of cell cycle-related proteins Cyclin A, Mdm2, C-MYC, etc., thus promoting cell proliferation [42,74,75]. MYC/MIZ1 inversely regulated the expression of circadian clock genes and affected the cell cycle and proliferation [76]. These results indicated that the circadian clock mainly regulates cell proliferation by blocking the cell cycle process or affecting the expression of cell cycle activators/cell cycle suppressors. Overexpression of the cell cycle activator or inhibitor changes the cell number but also leads to reverse changes in cell size [77]. In the Xenopus nervous system, inhibition of progenitor proliferation results in fewer but bigger neurons [78]. Knockdown of the *BmCry1* gene in BmN cells resulted in accelerated cell proliferation and a smaller nucleus, which may be related to cell cycle changes; however, we were more concerned about the metabolic changes of Cry1-KD.

Many cells, ranging from microbes to mammals, use aerobic glycolysis during rapid proliferation, suggesting that it may play a fundamental role in supporting cell growth [79]. In healthy cells, the PI3K/AKT/mTOR signaling pathway stimulates glucose input and glycolysis, promoting cell proliferation [80,81]. AMPK inhibits cell proliferation by inhibiting anabolic and catabolic pathways [82]. BmN cells with knockdown of the *BmPer* gene had inhibited glucose metabolism and slowed cell proliferation [34]. In the study of cancer cells, it is also found that the changes in glucose metabolism are always positively correlated with cell proliferation [83]. Cannabinoid receptor 2 (CB2R) inhibits glycolysis by downregulating HIF-1α, thereby inhibiting the proliferation of mouse liver macrophages [84]. Inhibition of genes related to glucose metabolism *OIP5*, *SMC4*, and *NUP107* also inhibited cell cycle progression [85]. These studies indicated that glucose metabolism is an important target for regulating cell proliferation. The proliferation of Cry1-KD cells was accelerated, which may be related to the upregulation of glucose metabolism.

### 3.3. Metabolic Difference between Knockdown BmCry1 and BmPer in BmN Cells

Previously, we reported the impact of the knockdown of *BmPer* on the metabolism of BmN cells [34]. We also investigated the reason that BmN cell proliferation slowed down after the knockdown of the *BmPer* gene [43]. In contrast to slowed growth of Per-KD cells, *BmCry1* knockdown significantly increased cell proliferation. We focused on metabolomics data and summarized the differences in the impacts of knockout of *BmCry1* and *BmPer* on BmN cell growth and metabolism (Appendix A). The results showed that carbohydrates and most organic acids and amino acids involved in the metabolism of glucose and lipids showed opposite trends. The overall level of glycometabolism was upregulated in Cry1-KD, whereas they were upregulated in Per-KD cells.

Even though PER and CRY both play important roles in inhibiting expression in the feedback loop of circadian clocks, PER protein protects the phosphorylation of CLOCK proteins by preventing damage from CRY proteins [86]; this is closely related to the entire clock oscillation and transcriptional rhythm of the output system [87]. Previous studies also showed that PER proteins have an opposite mechanism to CRY proteins, which is different from the traditional model, wherein PER proteins are transcriptional repressors of *Per* and *Cry* [88,89,90]. In addition, studies also showed that PER2 may have a positive effect on transcription in a promoter-specific manner [88,91]. Moreover, at certain stages and tissues, PER may play an antagonistic role with CRY [92,93]. It is therefore reasonable to conclude that CRY and PER proteins may play an antagonistic role in affecting metabolism in BmN cells. 

In zebrafish, computer simulations have demonstrated that many purine and pyrimidine metabolites exhibit the same rhythm phase, and de novo purine synthesis pathways have a great impact on the cell cycle [16]. In *Danaus plexippus*, the CRY1 protein transmits photo signals to PER and CRY2 proteins; these proteins then further regulate the transcription level genes by inhibiting CLK/CYC transcriptional activation, thereby implementing signal output [5]. Therefore, we conclude that the knockdown of *BmCry1* and *BmPer* may partially relieve the transcriptional repression of clock-controlled genes. Our study found that both Cry-KD and Per-KD cell lines showed similar changes in regard to decreased cell nuclei diameters and decreased cell volume, as well as upregulation of nucleotide metabolism levels. We were unable to explain why the nuclei were smaller after the knockdown of *BmPer* and *BmCry*, or why the levels of nucleotide and purine metabolites were upregulated. We suppose that these changes may be related to changes in the transcriptional repression of clock-controlled genes.

## 4. Materials and Methods

### 4.1. Preparation of the BmCry1 Knockdown (Cry1-KD) Cells 

The *B. mori* ovary cells (BmN) were cultured in Grace’s insect medium (Corning, NY, USA) with 10% FBS (Corning, NY, USA) at 26 °C in constant darkness (DD).

Pre-microRNA with mir-30 was used to construct transfection plasmid short hairpin RNAs (shRNAs). We designed three interference sites of *BmCry1*, and the mRNA levels of *BmCry1* were detected by qRT-PCR after transfection. We found that the Cry1-shRNA3 had a knockdown efficiency of 73% for *BmCry1* transcript levels, while Cry1-shRNA1, Cry1-shRNA2 or Cry1-shRNA1 + 3 had a lower knockdown efficiency (50%) (Appendix A). The *BmCry1* cDNA sequence ^1856^AAGAACGTGCCAACTGTATAA^1876^ was chosen for siRNA *Cry1* and inserted into shRNA to obtain shRNA-Cry1 (GCGACTTATACAGTTGGCACGTTCTTCTGTGAAGCCACAGATGGGAAGAACGTGCCAACTGTATAAGCTGC), which was then inserted into the OpIE2 promotor driven pIZT/V5-His/Cat vector between SacI^578^ and SpeI^651^ to generate the shRNA-Cry1 expression vector (Appendix A). The preparation of *BmCry1* knockdown cells was performed as described previously [34]. Briefly, BmN cells were cultured in Grace’s Insect Medium with 10% FBS at 26 °C in DD. We diluted 2 μL of Lipofectamine LTX reagent and 0.5 mg DNA plus reagent vector in 25 μL of Grace’s Insect Medium and mixed after a 10 min incubation. The mixture was then used to transfect BmN cells in 24-well plates for 48 h, followed by the addition of 100 μg/mL bleomycin. The transfection efficiency was counted by fluorescence microscopy. The shRNA-Cry1 expression vector transfected BmN cells for 2 days, and the number of green-labeled cells was about 50% (Appendix A). The medium was changed every 48 h and the bleomycin concentration in the medium was maintained at 100 μg/mL. The interference plasmid was continuously transfected, cells were screened with bleomycin for 2 weeks, and the number of green-labeled cells was about 90% or more (Appendix A). The efficiency of shRNA-mediated *BmCry1* knockdown was determined using qRT-PCR and Western blotting. Prior to qRT-PCR and Western blotting, both cell lines were synchronized using Dex (0.1 μM) for 2 h in darkness. The BmN cells were identified as *BmCry1* knockdown cells (Cry1-KD) after stable interference of the *BmCry1* gene, and cultured in the medium containing 100 μg/mL bleomycin at 26 °C in DD.

### 4.2. Cell Proliferation Assay

The 3–(4, 5)–dimethylthiazo (–z–y1) -3, 5-diphenytetrazoliumbromide (MTT) method was used to determine cell growth, according to the manufacturer’s instructions (C0009, Beyotime, Nantong, Jiangsu, China). Briefly, 100 μL of cell solution (~2 × 10^3^ cells) and 10 μL of MTT (5 mg/mL) solution were added to each well of a multi-well plate and incubated for 4 h at 37 °C. Formazan solution (100 μL) was then added and the mixture was incubated for a further 4 h at 37 °C. The absorbance at 570 nm was then measured using a microplate reader (Eon, Biotek, Winooski, VT, USA). All experiments were performed in triplicates.

### 4.3. Cell Staining

WT or Cry1-KD BmN cells were inoculated into culture plates for 12 h, washed with 1 mL PBS, and then fixed with 4% paraformaldehyde for 15 min. They were further treated with 0.1% Triton X-100 and stained with Dil staining solution (Beyotime, Nantong, Jiangsu, China) for 1 min. Finally, cells were stained with 4′–6-diamidino–2–phenylindole (DAPI; Beyotime) for 5 min at room temperature. It is important to ensure that PBST (PBS with 0.05% Tween-20) is used to fully wash after each fixation, permeabilization, and staining. After staining, the cell nuclear size (the average value of the longest and shortest diameters) was surveyed promptly using Image-Pro Plus software on a fluorescence microscope (Olympus BX51, Tokyo, Japan). Five independent fields were analyzed for each plate, and all experiments were performed in triplicate.

### 4.4. Determination of Metabolomics

According to the reported method [94], 1 × 10^7^ cells were counted, then obtained by quenching and centrifugation at 1000× *g* for 1 min. Based on the method reported by Tao et al. (2017), cell samples to be determined with GC/LC-MS were pretreated. Samples were resuspended in 500 μL of methanol, precooled to −80 °C and centrifuged twice (15,000× *g* for 1 min). The supernatant was further resuspended in ultrapure water, frozen in liquid nitrogen, and centrifuged (15,000× *g*, 1 min). The supernatant of the treated samples was vacuumed and dried at 30 °C to generate the specimen.

Spectroscopic parameters of GC-MS (Agilent 7890A/5975C, Agilent, Santa Clara, CA, USA) and LC-MS (Waters UPLC, Waters, Milford, MA, USA) are shown in Appendix A, respectively. An HP-5 MS capillary column (5% phenyl methyl silox: 30 μm × 250 μm internal diameter, 0.25 μm; Agilent J &W Scientific, Folsom, CA, USA) was used for GC-MS, and a C18 column (1.7 μm, 2.1 × 100 mm; (BEH, Waters, Milford, MA, USA) was used for LC-MS. Metabolites from LC-MS detection were further confirmed using secondary MS. We first confirmed and obtained the empirical formula of the metabolites based on the exact molecular weight (molecular weight error < 30 ppm). We further used the exact molecular weight according to MS/MS fragment patterns to search and confirm potential biomarkers in the Human Metabolome Database http://www.hmdb.ca website (accessed on 15 October 2022), Metlin http://metlin.scripps.edu/website (accessed on 27 October 2022), massbank http://www.massbank.jp/ (accessed on 5 November 2022) and LipidMaps http://www.lipidmaps.org (accessed on 8 December 2022) databases. A total of 20 μL of the sample was taken from all tested samples to correct errors that may have occurred during the tests. GC-MS and LC-MS assays were carried out at BioNovoGene Co., Ltd. (Suzhou, China).

### 4.5. qRT-PCR Analysis

WT and Cry1-KD cells were sampled every 4 h for 24 h for qRT-PCR. Total RNA was isolated with RNAiso™ Plus (TaKaRa, Dalian, China) from WT and Cry1-KD BmN cells, and cDNA was synthesized with the PrimeScript RT (Perfect Real Time) Reagent Kit with gDNA Eraser (TaKaRa), according to the manufacturer’s instructions. All reactions were carried out in a total reaction volume of 20 μL using an ABI StepOnePlus™ PCR system (Ambion, Foster City, CA, USA) and the fluorescent dye SYBR Premix Ex Taq (TaKaRa). Transcript levels of *BmCry2*, *BmPer*, *BmTim*, *BmClk*, and *BmCyc* were obtained under the following reaction conditions: 95 °C for 30 s, then 40 cycles at 95 °C for 5 s and 60 °C for 30 s. After PCR, we used a melting curve analysis to confirm the amplification of the specific products. The data were normalized with endogenous *BmRp49*. All the experiments were performed in triplicate. The gene-specific primers used in this study are shown in Appendix A.

### 4.6. Determination of Enzyme Activity

Soluble proteins from WT and Cry1-KD cells were extracted for enzyme activity assay. As mentioned in Tao et al. (2017), enzyme activities of HK, PFK, and PK were determined with the corresponding enzyme activity assay kit (Jiancheng, Nanjing, China), with glucose, glucose-6-phosphate, and phosphoenolpyruvate as substrates, respectively, according to the manufacturer’s instructions. One unit of HK, PFK, and PK, respectively, was defined as the consumption of 1 nM nicotinamide adenine dinucleotide/min/mg of cell protein (U/mg protein). Protein concentrations were measured with a BCA kit (Beyotime).

### 4.7. Western Blotting

Proteins were extracted from WT and Cry1-KD cells, and concentrations were measured with a BCA kit (Beyotime). Western blotting was performed according to the method described by Tao et al. (2017). Resultant protein bands were quantified using a ChemiDoc Touch Imaging System (Bio-Rad, Hercules, CA, USA). The BmCRY1 protein amino acid sequence was acquired from the NCBI (GenBank accession number ADM86934). After peptide sequence design, synthesis, and purification, the peptide sequence of the CRY1 protein, NH2-RLDPSGEYVRRYVPECCONH2, was used as an antigen to immunize New Zealand rabbits. The antigenic epitopes used for raising the polyclonal antibodies for CRY1 were at the amino acid residues of 432–446. The NH2 at the N-terminal and the CCONH2 at the C-terminal were used to conjugate with the carrier protein KLH. An enzyme-linked immunosorbent assay (ELISA) was used to detect the antibody titer. The ELISA result was about 3 when they were diluted 1000 times. As the titers of the antisera were more than 1, we affirmed that they were efficient antibodies. The antibodies were purified using affinity purification. All of these procedures were completed by Abgent Biotechnology Co., Ltd. (Suzhou, China).

### 4.8. Immunohistochemistry

WT or Cry1-KD BmN cells were inoculated in culture plates for 12 h and washed with 1 mL phosphate-buffered saline (PBS). Cells were fixed with 4% paraformaldehyde solution for 15 min at room temperature and washed three times with PBS. Then, the cells were treated with permeabilization buffer (0.1% sodium deoxycholate and 2% Tween-20 in PBS solution) for 30 min at room temperature and washed three times with PBS. The blocking solution was added and shaken for 1 h at 37 °C. Cells were incubated with CRY1 polyclonal antibody for *Bombyx mori*, and diluted 1: 100 with TBS (PBS with 0.05% Tween-20) at 4 °C for 12 h. After further washing three times with TBS, cells were then added to 500 μL TBS and 0.5 μL goat anti-rat IgG(H + L) FITC (MultiSciences, Hangzhou, China) and incubated at 37 °C for 2 h in the dark. Cells were washed three times again with TBS and observed using a fluorescence microscope (Olympus BX51, Tokyo, Japan).

### 4.9. Statistical Analysis

Raw data from GC-MS and LC-MS were processed as described in Tao et al. (2017), using XCMS www.bioconductor.org/ (accessed on 27 October 2022). We used a partial least-squares discriminant analysis (PLS-DA), first principal component variable importance in projection (VIP) values (VIP ≥ 1), and Student’s *t*-test (*p* < 0.05) to screen the differential metabolites. Analysis of the above data as well as PCA and heat map functions were all implemented using R 3.0.3 www.r-project.org (accessed on 22 December 2022).

Metabolite association analysis and statistical tests using Pearson’s correlation coefficients with cor() and cor.test() functions were analyzed using the R package. The metabolite correlation threshold was set to 0.8, and a false discovery rate of *p* ≤ 0.05 was considered significant; the relationships among metabolites for constructing the metabolic pathways were based on the Kyoto Encyclopedia of Genes and Genomes (KEGG) database http://www.genome.jp/kegg/ (accessed on 27 December 2022). A univariate analysis of variance (ANOVA) was used to determine the significance of differences in relative contents between different groups. Pathway activity profiling (PAPi) was used to predict and compare the relative activity of different metabolic pathways during different comparisons. Throughout the analysis process, data normalization and related statistical analysis were carried out in MetaboAnalyst https://www.metaboanalyst.ca (accessed on 14 January 2023). Enzyme activity and gene expression levels were analyzed by a univariate ANOVA. 

## Figures and Tables

**Figure 1 ijms-24-05435-f001:**
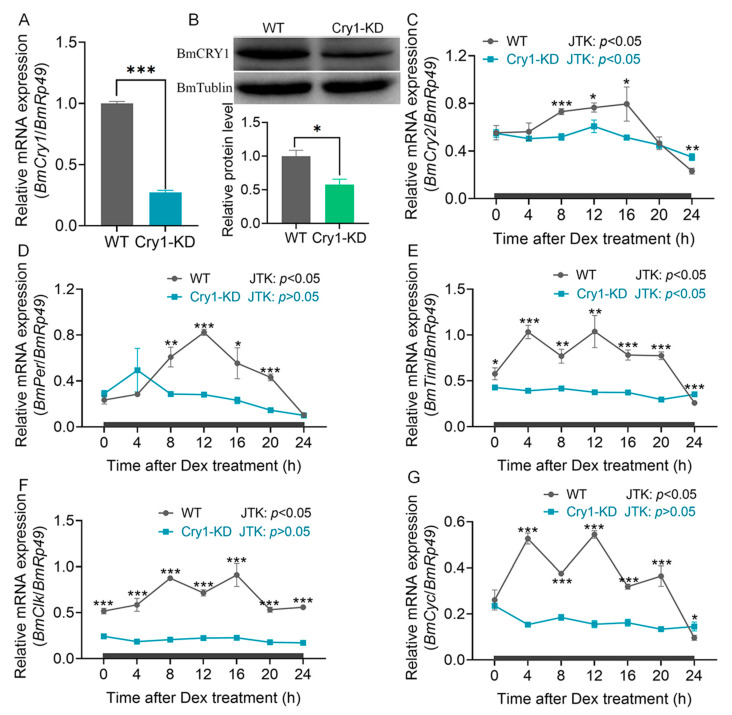
*BmCry1* knockdown efficiency and clock gene expression. (**A**) *BmCry1* transcripts were analyzed by qRT-PCR with *BmRp49* as the internal control (*n* = 3). (**B**) BmCRY1 protein levels with BmTUBLIN protein as the internal control (*n* = 2). (**C**–**G**) Transcripts of the five clock genes were analyzed with qRT-PCR using *BmRp49* as the internal control (*n* = 3). Cry1-KD and WT cells were cultured at 26 °C in DD. The cells were treated with 100 μM dexamethasone (DEX) for 2 h, and the transcripts of these genes were investigated. Samples were analyzed at 4 h intervals for 24 h. JTK: *p* < 0.05 indicated that the gene expression had a 24- h circadian rhythm, JTK: *p* > 0.05 indicated no 24 h circadian rhythm in gene expression. The black lines indicate constant darkness. WT, wild-type cells; Cry1-KD, *BmCry1* knockdown cells. *, *p* ≤ 0.05; **, *p* ≤ 0.01; ***, *p* ≤ 0.001.

**Figure 2 ijms-24-05435-f002:**
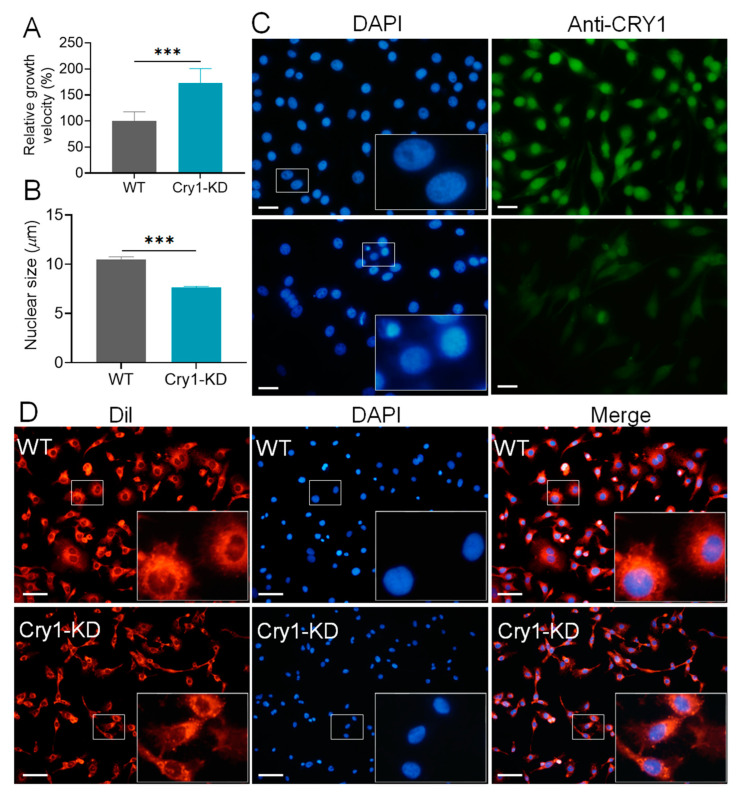
Effect of knockdown *BmCry1* on cell growth. (**A**) Cell growth velocity was measured using the MTT method (*n* = 3). (**B**) Cell nuclei diameters differ between Cry1-KD cells and WT cells. When the cell fusion degree in cell vials reached 70%, cells were seeded on coverslips. DAPI staining was performed on the cells after adherence (at least 12 h) and the diameters of the nuclei were measured. cell nuclei diameters were quantified by analyzing cells in five randomly selected fields on a coverslip (*n* = 3). (**C**) BmCRY1 protein content in BmN and Cry1-KD cells was investigated by immunofluorescence. bar = 50 μm. (**D**) Cell nuclei stained with DAPI and cell membranes stained with Dil. bar = 50 μm. WT, wild-type cells; Cry1-KD, *BmCry1* knockdown cells. ***, *p* ≤ 0.001.

**Figure 3 ijms-24-05435-f003:**
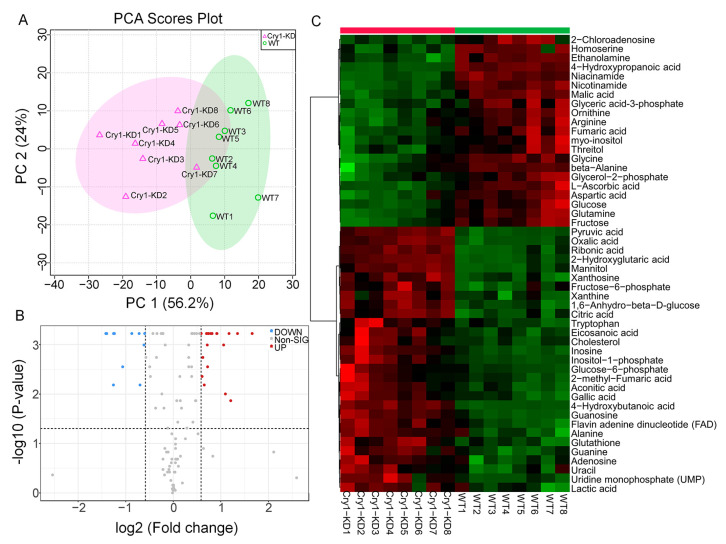
Differential metabolites based on LC-MS and GC-MS/MS metabolomics. (**A**) Principal component analysis; (**B**) volcano plot. The differential metabolites were screened out under the condition of fold change > 1.5 and FDR < 0.05; (**C**) heatmap of the top 50 differential metabolites. WT, wild-type cells; Cry1-KD, *BmCry1* knockdown cells.

**Figure 4 ijms-24-05435-f004:**
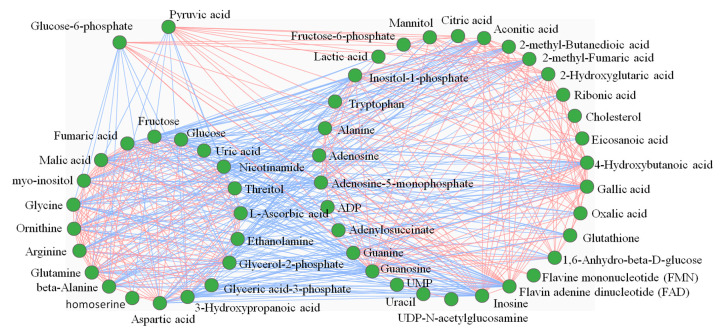
The correlation network diagram of differential metabolites. The Pearson correlation coefficient was used for metabolite correlation analysis using the calculation method cor () function in the R language package www.r-project.org (accessed on 22 December 2022). The Pearson correlation coefficient threshold was set to 0.8. The red lines represent positive correlations between substances, and blue lines represent negative correlations between substances.

**Figure 5 ijms-24-05435-f005:**
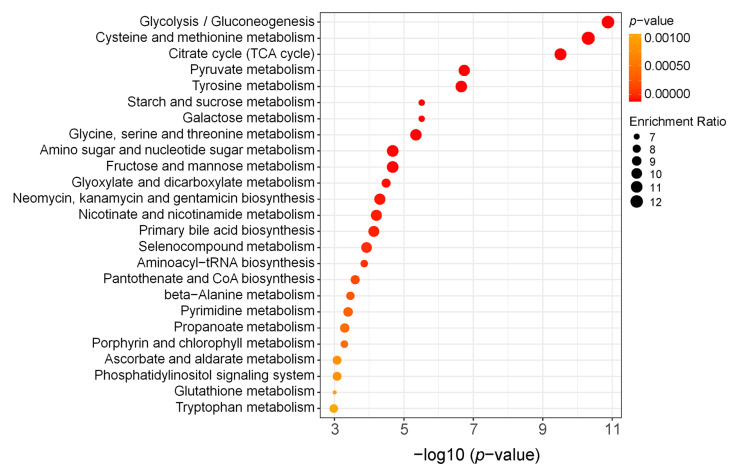
KEGG-enrichment analysis of differential metabolites. The top 25 metabolic pathways with the most significant differences were selected. The size of the dot represents the number of enriched genes; the color of the dot represents the *p* value.

**Figure 6 ijms-24-05435-f006:**
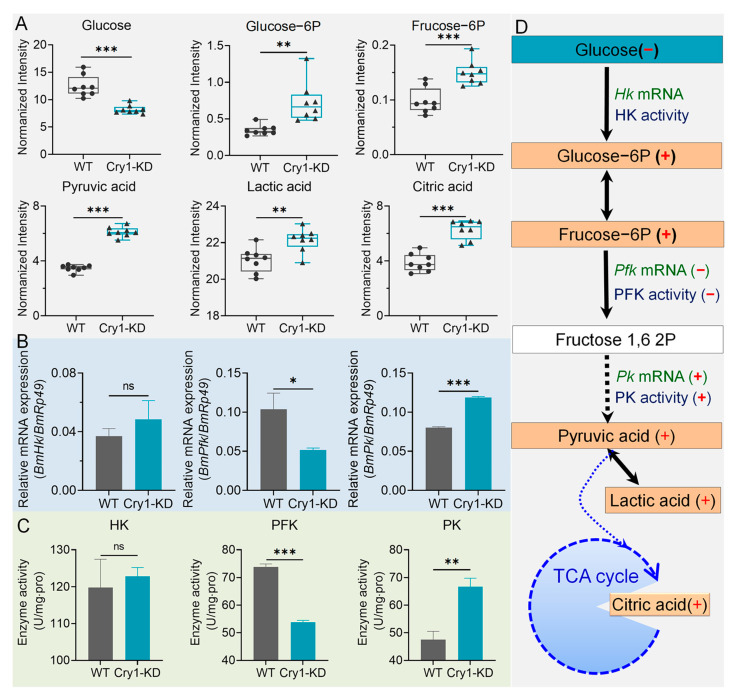
Effect of *BmCry1* knockdown on glucose metabolism in BmN cells. (**A**) Changes in the contents of main substrates and products associated with glucose metabolism are based on metabolomic measurements (*n* = 8). Black dots represent WT cells; black triangles represent *BmCry1* knockdown cells. (**B**) Transcription levels of the main rate-limiting enzyme-encoding genes (*n* = 3). (**C**) Activities of the main rate-limiting enzymes (*n* = 3). (**D**) Glucose metabolic pathways were affected by *BmCry1* knockdown. WT, wild-type cells; Cry1-KD, *BmCry1* knockdown cells. HK, hexokinase; PFK, phosphofructokinase; PK, pyruvate kinase; +, increased content or enzymatic activity or downregulated transcription of coding gene; −, decreased content or enzyme activity or downregulated transcription of the coding gene. *, *p* ≤ 0.05; **, *p* ≤ 0.01; ***, *p* ≤ 0.001; ns, non-significant.

## Data Availability

Data sharing not applicable.

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
