# Peer review of "Inhibition of Expression of the Circadian Clock Gene Cryptochrome 1 Causes Abnormal Glucometabolic and Cell Growth in Bombyx mori Cells"

_ijms, 2023, doi:10.3390/ijms24065435_

Round 1

Reviewer 1 Report

This study represents a thorough and interesting analysis of the effect of BmCry 1 knockdown and demonstrates loss of clock function and important metabolic changes related to loss of Cry1. The phenotype is striking in that it is very different from the Drosophila Cry mutant which has limited or no effect on cell growth. Some minor comments are concerning the english language - eg. Blu-Ray (page one) is a term that does not exist (presumably mean blue light) and there are other grammatical and text problems, it would benefit by an editor going through it. Another problem concerns the cell culture conditions, it is no where discussed whether they are on a dark light cycle or if there is any light response. In Dm the Cry1 is involved in inputting the light signal to the clock and binds directly to tim and per in a light dependent manner, the authors should at least discuss the light/dark cycle they used and whether they expect a change if cells are kept under DD or LL. Another minor point concerns comparison with Cry2 - in butterfly and vertebrates there is only Cry2 which seems to be doing all of the things that Cry1 is doing in this study (ie maintaining clock function). The authors should speculate on what they think Cry2 is doing in this system, perhaps providing a bit more background on what is known about other insects that have both genes (cry1 and cry2).

Overall a very good and valuable study.

Reviewer 2 Report

The manuscript by Qiu et al. examined the role of the circadian clock gene Cryptochrome 1 (Cry 1) in the body or cell metabolism of the silkworm ovary cell line (BmN). They found that Cry 1 depletion causes abnormal glucometabolic and cell growth in BmN. The major conclusions of this research are justified by the results. The methodology seems to be correct in most experiments and the results of this work may be worth publishing. However, the study requires improvement in some aspects. Please consider the following points:

Major revision:

1. Cry1 depletion accelerated cell growth, why did not the authors examine the signaling of cell proliferation, such as cell cycle? And I believed that the authors did not clearly describe the relationship between cell proliferation and glucometabolic in the part of Introduction/Discussion, they just introduced/discussed the effect of circadian clocks on metabolism, so the authors need to add this information.

2. In line 78, the knockdown efficiency (47%) of Cry 1 shRNA on the level of CRY protein was not enough, the authors should improve this or I think that the cDNA sequence of Cry 1 was not the best and two shRNA vectors may be needed to further confirm your results.

3. The shRNA-Cry1expression vector was transfected into BmN cells, how did the authors examine the transfection efficiency? In addition to PCR and WB, immunofluorescent staining was also needed in figure 2C, it may make your paper convincing that knockdown of Cry 1 made a smaller nuclei.  

Round 2

Reviewer 2 Report

No